# Explicit and Effectively Symmetric Schemes for Neural SDEs

## Abstract

Backpropagation through (neural) SDE solvers is traditionally approached in two ways: discretise-then-optimise, which offers accurate gradients but incurs prohibitive memory costs due to storing the full computational graph (even when mitigated by checkpointing); and optimise-then-discretise, which achieves constant memory cost by solving an auxiliary backward SDE, but suffers from slower evaluation and gradient approximation errors. Algebraically reversible solvers promise both memory efficiency and gradient accuracy, yet existing methods such as the Reversible Heun scheme are often unstable under complex models and large step sizes. We address these limitations by introducing a novel class of stable, near-reversible Runge–Kutta schemes for neural SDEs. These *Explicit and Effectively Symmetric (EES)* schemes retain the benefits of reversible solvers while overcoming their instability, enabling memory-efficient training without severe restrictions on step size or model complexity. Through numerical experiments, we demonstrate the superior stability and reliability of our schemes, establishing them as a practical foundation for scalable and accurate training of neural SDEs.

## 1 Introduction

*Neural stochastic differential equations (SDEs)* have recently emerged as a flexible tool for modelling stochastic dynamics, with training typically cast as a distribution-matching problem between generated and observed trajectories. Several approaches have been proposed in the literature, differing mainly in the choice of discriminating divergence. SDE-GANs (Kidger et al., 2021a) use the 1-Wasserstein distance, while Latent SDEs (Li et al., 2020) optimize with respect to the KL divergence via variational inference, and can be viewed as variational autoencoders. Another alternative proposed by Issa et al. (2023) trains neural SDEs non-adversarially using maximum mean discrepancies (MMD) with *signature kernels* (Király & Oberhauser, 2019; Salvi et al., 2021a; Lemercier et al., 2024), a recently introduced family of kernels on path space which received significant attention due to their efficiency in handling path-dependent problems (Salvi et al., 2021b; Lemercier et al., 2021; Pannier & Salvi, 2024; Muça Cirone & Salvi, 2025b).

While effective, the stochastic calculus underpinning SDEs can be technically cumbersome, especially when developing higher order solvers, deriving and analysing backpropagation algorithms, or extending to rougher noises than Brownian motion. A natural way to bypass these limitations is to view SDEs through the lens of *rough paths*. Rough path theory (Lyons, 1998; Gubinelli, 2004; 2010) provides a deterministic calculus on path space that extends classical Itô integration beyond semimartingales, enabling one to treat SDEs as a special case of *rough differential equations (RDEs)*

$$dy(t) \; = \; f_0(t, y_t) \, dt \; + \; f(t, y_t) \, d\mathbf{X}(t), \qquad y_{t_0} = y_0 \in \mathbb{R}^m, \tag{1}$$

where $\mathbf{X}$ is a *rough path* above a driving signal $X = (X^1, ..., X^d) : [0, T] \to \mathbb{R}^d$, $f = (f_1, ..., f_d)$ and $f_i : \mathbb{R}^m \to \mathbb{R}^m$ are vector fields for $i = 0, ..., d$ which determine the system dynamics. In fact, there is a large class of stochastic processes which can be "naturally lifted" to rough paths, including Gaussian processes such as fractional Brownian motion with Hurst parameter $H > 1/4$ and Volterra processes (Friz & Victoir, 2010). Viewing SDEs through the lens of RDEs provides a significant conceptual simplification. In the RDE framework, the driving signal $\mathbf{X}$ is treated as a rough path, which abstracts away the stochastic integral and allows one to work with deterministic calculus on path space. Rough path theory has become a powerful mathematical framework for analyzing modern machine learning models. It has provided the foundation for proving theoretical properties of neural

differential equations (Morrill et al., 2021; Arribas et al., 2020; Cirone et al., 2023; Holberg & Salvi, 2024), sequence-to-sequence architectures (Kidger et al., 2019), and more recently for deep selective state-space models (SSMs) (Muça Cirone et al., 2024; Muça Cirone & Salvi, 2025a; Walker et al., 2025) and score-based diffusion models (Barancikova et al., 2024). for a survey of recent applications of rough path theory to machine learning, we refer the reader to (Fermanian et al., 2023).

From an algorithmic perspective, the RDE formulation also plays a central role in training neural SDEs (NSDEs), where one must perform backpropagation through an SDE solver. For example, the derivation of the *adjoint method*—a key tool for backpropagation through differential equation solvers—becomes far more transparent when formulated in terms of RDEs, avoiding much of the technical machinery required in the stochastic setting; see for instance (Cass & Salvi, 2024). More generally, the RDE viewpoint unifies backpropagation across ODEs, SDEs, and controlled differential equations (CDEs) (Kidger et al., 2020), offering a clean pathwise calculus that is both mathematically rigorous and practically aligned with autodiff frameworks. Several strategies have been proposed. A first approach, known as *discretise-then-optimise*, directly backpropagates through the solver's internal operations. This yields accurate gradients and is computationally efficient, but requires storing all intermediate states, making it memory-intensive. A second approach, *optimise-then-discretise*, instead derives a backwards-in-time adjoint equation and solves it numerically using another call to the solver. This eliminates the need to store intermediate quantities, resulting in constant memory cost with respect to solver depth. However, it typically produces less accurate gradients and is slower due to the need to recompute forward trajectories during the backward pass.

A third option leverages *algebraically reversible solvers*, which enable the exact reconstruction of the solution trajectory of a differential equation from its terminal point, allowing for accurate and memory-efficient backpropagation. In the setting of an autonomous ODE

$$dy_t = f(y_t)dt, \tag{2}$$

a one step method $y_{n+1} = y_n + \Phi_h(y_n)$ is said to be *reversible* or *symmetric* if a step of the method starting from $y_1$ with a negative step size exactly recovers the initial condition $y_0$, that is, $\Phi_{-h} = \Phi_h^{-1}$. Whilst reversible schemes offer an efficient approach to backpropagation through differential equations, such schemes are difficult to construct. It is well known that Runge–Kutta schemes are reversible only if they are implicit, making them unsuitable for applications to Neural ODEs. More generally, symmetric parasitism-free general linear methods cannot be explicit (Butcher et al., 2016). To overcome this problem, existing reversible methods proposed in literature, such as the asynchronous leapfrog integrator (ALF) (Zhuang et al., 2021) and the Reversible Heun method (Kidger et al., 2021b), track auxiliary states as part of the integration. Whilst this technique allows for explicit reversible schemes, it comes at the price of low stability. Both ALF and Reversible Heun are well-known to be unstable, and fail when integrating complex equations. For example, Zhang & Chen (2021) find that Reversible Heun is too unstable for their applications, stating:

> *"Regardless the accuracy and memory advantages of Reversible Heun claimed by torchsde, we found this integration approach is less stable compared with simple Euler integration without adjoint and results in numerical issues occasionally. We empirically found that methods without adjoint are more stable and lower loss compared with adjoint ones, even in low dimensional data"* *(Zhang & Chen, 2021)*

A potential solution to the difficulties of reversible solvers comes with the class of *Explicit and Effectively Symmetric (EES)* Runge–Kutta schemes introduced in Shmelev et al. (2025). Instead of considering exactly reversible schemes, the authors propose schemes which are almost reversible, up to an acceptable tolerance level. Such schemes offer stable, explicit integration methods which are virtually indistinguishable from truly symmetric schemes in practice. The authors demonstrate the efficacy of EES schemes on classical ODEs, and show that EES schemes are capable of producing similar results to classical schemes such as RK3 and RK4. In Shmelev et al. (2025), the formulation of EES schemes is limited to ODEs, without an explicit generalisation to the case of SDEs or RDEs.

Derivative-free *Runge–Kutta (RK) methods* for RDEs were introduced by Redmann & Riedel (2022). Similarly to classical RK schemes for ODEs, the study of these methods is conducted through the formalism of B-series (Hairer et al., 2006; McLachlan et al., 2015; Butcher, 2021). A B-series is an infinite series representation of a method, indexed by (labelled) non-planar rooted trees. A brief overview of B-series is given in Appendix A. A natural consequence of this analysis is that a general

Runge–Kutta method for RDEs is given in terms of tree-iterated integrals of the underlying driving process. In practice, these tree-iterated integrals cannot be simulated directly as their distributions are often intractable. Following (Deya et al., 2012), Redmann & Riedel (2022) replaced these tree-iterated integrals with products of increments of the driving path. This substitution simplifies the derivation of Runge–Kutta coefficients and makes it feasible to establish order conditions up to any desired order.

In this paper, we present a formulation of EES Runge–Kutta schemes for RDEs, and demonstrate their efficacy as integrators for Neural SDEs. The paper is structured as follows. Section 2 gives an overview of existing reversible methods for neural differential equations. Section 3 recounts the framework of Runge–Kutta methods for RDEs introduced in Redmann & Riedel (2020). Section 3 begins by introducing EES schemes for ODEs. Using the framework of Redmann & Riedel (2020), we extend EES schemes to the case of RDEs, and derive results regarding their orders of convergence. Through *mean-square stability* analysis, we show that EES schemes possess a similar stability domain to classical RK3 and RK4 schemes when applied to SDEs, and are significantly more stable than existing reversible schemes designed for Neural SDEs. We end Section 3 by outlining a backpropagation algorithm for explicit RDE Runge–Kutta solvers such as EES schemes. In Section 4 we demonstrate the efficacy of EES schemes as integrators for Neural SDEs with two experiments concerned with the learning of extreme stochastic dynamics where existing reversible solvers fail due to instability, and show that EES successfully overcomes this issue. Section 5 closes this work with a summary of the results, limitations and potential avenues for future work.

## 2    EXISTING REVERSIBLE SDE SOLVERS

The major drawback of classical reversible schemes is their low efficiency. It is well known that Runge–Kutta schemes are reversible only if they are implicit. More generally, symmetric parasitism-free general linear methods cannot be explicit (Butcher et al., 2016). A limited number of efficient reversible solvers have been proposed in the literature. The asynchronous leapfrog integrator (ALF) (Zhuang et al., 2021) for Neural ODEs overcomes the barrier of implicit schemes by tracking an additional state $v$ as part of the integration. Applied to the ODE equation 2, the update rule of the ALF scheme can be written as

$$y_{n+2} = y_n + hf\left(t_n + \frac{h}{2}, y_n + \frac{h}{2}v_n\right),$$

$$v_{n+2} = 2f\left(t_n + \frac{h}{2}, y_n + \frac{h}{2}v_n\right) - v_n.$$

A similar approach is taken for SDEs of the form

$$dy_t = g(t, y_t)dt + f(t, y_t)dW_t \tag{3}$$

by the Reversible Heun method (Kidger et al., 2021b), where the integration step reads

$$y_{n+1} = y_n + \frac{1}{2}(g(t_n, v_n) + g(t_{n+1}, v_{n+1}))\Delta t + \frac{1}{2}(f(t_n, v_n) + f(t_{n+1}, v_{n+1}))\Delta W_n,$$

$$v_{n+1} = 2y_n - v_n + g(t_n, v_n)\Delta t + f(t_n, v_n)\Delta W_n.$$

The Reversible Heun method is efficient, requiring only one evaluation of the drift $g$ and the diffusion $f$ per step. However, the method is inherently unstable.

*Theorem* 2.1. (Kidger et al., 2021b, Theorem D.19) Suppose that the Reversible Heun method is used to obtain a solution $\{y_n, v_n\}_{n \geq 0}$ to the linear test ODE $dy = \lambda y\, dt$, where $\lambda \in \mathbb{C}$ and $y_0 \neq 0$. Then $\{y_n, v_n\}_{n \geq 0}$ is bounded if and only if $\lambda h \in [-i, i]$.

As remarked in Kidger et al. (2021b), this domain is also the absolute stability region for the reversible asynchronous leapfrog integrator (Zhuang et al., 2021). This instability has proven to be a significant bottleneck in certain practical applications. In McCallum & Foster (2024), a method was proposed for transforming any ODE integration method $y_{n+1} = y_n + \Psi_h(t, y_n)$ into one which is reversible, by taking the update

$$y_{n+1} = \lambda y_n + (1 - \lambda)v_n + \Psi_h(t, v_n),$$

$$v_{n+1} = v_n - \Psi_{-h}(t_{n+1}, y_{n+1}),$$

for a given coupling parameter $\lambda \in (0, 1]$. The method offers a way of constructing reversible schemes with larger stability domains than those of the ALF and Reversible Heun integrators (McCallum & Foster, 2024, Theorem 2.3). However, the resulting stability domain of the transformed method is typically much smaller than that of the underlying method $\Psi$, and depends additionally on the coupling parameter $\lambda$.

## 3 EXPLICIT AND EFFECTIVELY SYMMETRIC (EES) SCHEMES FOR RDEs

We will adopt the general framework of RDEs for our analysis of reversible SDE solvers, following the work of Redmann & Riedel (2020). As discussed in the introduction, a generalised Runge–Kutta scheme for RDEs can be formulated in terms of tree-iterated integrals of the underlying driving rough path. For a brief introduction to these generalised methods, we refer the reader to Appendix B. Computation of such tree-iterated integrals is usually not tractable in practice, and so we adopt the simplified scheme given instead in terms of products of increments of the driving rough path.

### 3.1 SIMPLIFIED RUNGE–KUTTA METHODS FOR RDEs

Throughout, we will consider rough differential equations (RDEs) of the form

$$dy_t = f(y_t)d\mathbf{X}_t, \tag{4}$$

where $\mathbf{X}$ is an $\alpha$-Hölder branched rough path for some $\alpha \in (0, 1]$ and $f$ is sufficiently smooth and bounded with bounded derivatives. For an introduction to branched rough paths, we refer the reader to Appendix A.3. Following Redmann & Riedel (2022), we assume that there exist smooth paths $\{X^h\}_{h>0}$ whose natural lifts to branched rough paths $\{\mathbf{X}^h\}_{h>0}$ converge (almost surely) to $\mathbf{X}$ under the metric for $\alpha$-Hölder rough paths $\varrho_\alpha$ (see Appendix A.3) as $h \to 0$. That is, $\mathbf{X}$ is a geometric rough path. Assume that this Wong-Zakai-type approximation converges at a rate $r_0 > 0$ with respect to the inhomogemeous rough path metric for geometric rough paths $\varrho_\alpha^g$, such that $\varrho_\alpha^g(\mathbf{X}^h, \mathbf{X}) = \mathcal{O}(h^{r_0})$. For examples of such convergence rates for Gaussian processes, see Friz & Riedel (2014). Let $y^h$ denote the solution associated with the driver $\mathbf{X}^h$,

$$dy_t^h = f(y_t^h)d\mathbf{X}_t^h. \tag{5}$$

A simplified Runge-Kutta scheme for $y^h$ is defined by

$$y_{n+1}^h = y_n^h + \sum_{m=1}^d \sum_{i=1}^s b_i f_m(k_i) X_{t_n, t_{n+1}}^{(m)}, \tag{6}$$

$$k_i = y_n^h + \sum_{m=1}^d \sum_{j=1}^s a_{ij} f_m(k_j) X_{t_n, t_{n+1}}^{(m)},$$

where $X_{t_n, t_{n+1}}$ denotes the increment of $X^h$ over $[t_n, t_{n+1}]$. For simplicity, we will assume equidistant grid points $t_{n+1} - t_n = h$ for all $n \geq 0$. The convergence rates derived in Redmann & Riedel (2020) for schemes of the form in equation 6 are given in Appendix B.1.

**Notation 3.1.** To avoid confusion, we will generally use $\Phi$ to refer to a classical ODE Runge–Kutta method, and $\Upsilon$ to refer to an RDE method of the form given in equation 6. Given an ODE Runge-Kutta scheme $\Phi$, we will write $\mathcal{R}(\Phi)$ to denote the RDE scheme of the form in equation 6 with the same coefficients $\{a_{ij}\}_{1 \leq i,j \leq s}$ and $\{b_i\}_{1 \leq i \leq s}$ as the ODE scheme.

### 3.2 EES SCHEMES FOR ODEs

EES schemes (Shmelev et al., 2025) are a class of Runge–Kutta methods which offer an efficient approach to reversible integration without compromising on stability. Given positive integers $m \geq n$, an explicit Runge–Kutta scheme $\Phi_h$ is said to be an $\mathrm{EES}(n, m)$ scheme if $\Phi_h$ is of order $n$ and applying the scheme $\Phi_{-h} \circ \Phi_h$ to an ODE recovers the initial condition up to order $m$. When $m$ is large, such schemes offer near-reversible behaviour, which is often sufficient for practical applications. Butcher tableaux for 3-stage $\mathrm{EES}(2, 5)$ and 4-stage $\mathrm{EES}(2, 7)$ schemes are derived in Shmelev et al. (2025). In Shmelev et al. (2025), the authors focus mostly on $\mathrm{EES}(2, 7)$ schemes, as these

significantly outperform $\mathrm{EES}(2,5)$ schemes as integrators for ODEs. For our applications to Neural SDEs, we will instead restrict ourselves to $\mathrm{EES}(2,5)$, as we do not expect the extra accuracy of $\mathrm{EES}(2,7)$ justifies the extra stage required in this case. Proposition 3.1 gives the general form of the Butcher tableau of a scheme belonging to $\mathrm{EES}(2,5)$ in terms of a parameter $x$, which we will denote by $\mathrm{EES}(2,5;x)$.

*Proposition* 3.1 ((Shmelev et al., 2025, Proposition 8.4)). 3-stage Runge–Kutta schemes belonging to $\mathrm{EES}(2,5)$ have a Butcher tableau of the form:

$$
\begin{array}{c|ccc}
0 \\
\dfrac{1+2x}{4(1-x)} & \dfrac{1+2x}{4(1-x)} \\
\dfrac{3}{4(1-x)} & \dfrac{(4x-1)^2}{4(x-1)(1-4x^2)} & \dfrac{1-x}{(1-4x^2)} \\
\hline
& x & \dfrac{1}{2} & \dfrac{1}{2}-x
\end{array}
\tag{7}
$$

for some $x \in \mathbb{R}$, $x \neq 1, \pm\frac{1}{2}$.

Recall from Theorem 2.1 that the stability domain of Reversible Heun is the interval $[-i, i]$, which is also the stability domain of the ALF integrator. The stability region for $\mathrm{EES}(2,5)$ schemes is significantly larger and is comparable to classical methods such as Kutta's RK4, as shown in Figure 1. Theorem 3.1 gives the exact form of this region for $\mathrm{EES}(2,5;x)$.

*Theorem* 3.1. Suppose that, for some $x \neq 1, \pm\frac{1}{2}$, $\mathrm{EES}(2,5;x)$ is used to obtain a solution $\{y_n\}_{n \geq 0}$ to the linear test equation $dy = \lambda y\, dt$, where $\lambda \in \mathbb{C}$ and $y_0 \neq 0$. Then $y_n \to 0$ as $n \to \infty$ if and only if

$$
\left| 1 + \rho + \frac{1}{2}\rho^2 + \frac{1}{8}\rho^3 \right| < 1,
$$

where $\rho = \lambda h$.

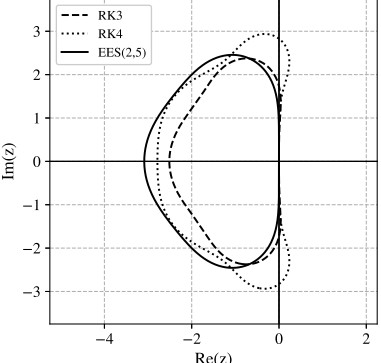

Figure 1: *Stability domain for* $\mathrm{EES}(2,5;1/10)$ *compared to Kutta's RK3 and RK4.*

The result follows from the fact that Runge–Kutta methods applied to linear test equations admit a linear update rule, $y_{n+1} = R(\rho)y_n$, where $R(\rho)$ is the stability function associated with the scheme. As such, the proof of Theorem 3.1 is simply a direct computation of the stability function $R(\rho) = 1 + \rho + \frac{1}{2}\rho^2 + \frac{1}{8}\rho^3$ for $\mathrm{EES}(2,5;x)$, which we omit here. We note that the stability region of $\mathrm{EES}(2,5;x)$ is completely independent of the choice of the parameter $x$. Motivated by the discussion in Shmelev et al. (2025, Section 8.1), we choose to fix the parameter $x = 1/10$ and refer to $\mathrm{EES}(2,5;1/10)$ as *the* $\mathrm{EES}(2,5)$ scheme from now on.

### 3.3 EES SCHEMES FOR RDES

We aim to define an analogue of EES schemes for RDEs and give convergence rates for the local and global errors of these schemes. To reason about the reversibility of schemes, it will be convenient to adopt the following notation.

**Notation 3.2.** Let $\{y_n^h\}_{i=0,\ldots,N}$ denote the solution of a Runge–Kutta scheme $y_{n+1}^h = y_n^h + \Upsilon(y_n^h, X_{t_n,t_{n+1}})$ of the form given in equation 6 applied to equation 5. Let $\bar{\Upsilon}$ denote the reverse-time scheme, $\bar{\Upsilon}(\cdot, X_{t_n,t_{n+1}}) := \Upsilon(\cdot, -X_{t_n,t_{n+1}})$, and let $\bar{y}_n^h$ denote the result of an n-fold application of $\bar{\Upsilon}$ to $y_n^h$, such that $\{\bar{y}_n^h\}_{i=0,\ldots,N}$ is a sequence of approximations to the initial condition $y_0$.

We may now define a class of EES schemes for RDEs, which we will denote by $\mathrm{EES}_{\mathcal{R}}$. In a similar fashion to Shmelev et al. (2025), we will define the schemes in terms of both the local error of the solution and the local error of the recovered initial condition when the scheme is run in reverse.

**Definition 3.1.** When applied to equation 5, a Runge–Kutta scheme $y_{n+1}^h = y_n^h + \Upsilon(y_n^h, \mathbf{X}_{t_n, t_{n+1}})$ of the form equation 6 is said to belong to $\text{EES}_{\mathcal{R}}(n, m)$ for $m \geq n$ if

$$y^h(t_1) - y_1^h = \mathcal{O}(h^{(n+1)\alpha}), \quad y_0 - \bar{y}_1^h = \mathcal{O}(h^{(m+1)\alpha}).$$

As a consequence of Redmann & Riedel (2020, Theorem 3.3), $\text{EES}_{\mathcal{R}}$ schemes can be constructed directly from EES schemes by using the same coefficients. A global error rate for the schemes is given below as a corollary of (Redmann & Riedel, 2020, Theorem 4.2), noting that the error in the recovery of $y_0$ from $\bar{y}_n^h$ is independent of the chosen Wong-Zakai approximation, and as such independent of $r_0$. Numerical experiments verifying the global rates on a test RDE driven by a fractional Brownian motion are presented in Appendix C.

*Theorem 3.2.* Let $\Phi \in \text{EES}(n, m)$ for some $m \geq n$. Then $\mathcal{R}(\Phi) \in \text{EES}_{\mathcal{R}}(n, m)$.

*Proof.* Since $\Phi$ is of order $n$, it follows from (Redmann & Riedel, 2020, Theorem 3.3) that $\mathcal{R}(\Phi)$ has a local error of order $(n + 1)\alpha$. Consider the scheme $\mathcal{R}(\bar{\Phi}) \circ \mathcal{R}(\Phi) = \mathcal{R}(\bar{\Phi} \circ \Phi)$. By definition of $\Phi$, the scheme $\bar{\Phi} \circ \Phi$ admits a B-series expansion $y_0 + \mathcal{O}(h^{m+1})$ (Shmelev et al., 2025). It follows from Redmann & Riedel (2020) that the corresponding B-series expansion for $\mathcal{R}(\bar{\Phi} \circ \Phi)$ is $y_0 + \mathcal{O}(h^{(m+1)\alpha})$. $\square$

*Theorem 3.3.* Let $\Upsilon \in \text{EES}_{\mathcal{R}}(n, m)$ for some $m \geq n$. Then

$$\max_{i=0,\ldots,N} |y(t_i) - y_i| = \mathcal{O}(h^{\eta_1}), \quad \max_{i=0,\ldots,N} |y_0 - \bar{y}_i| = \mathcal{O}(h^{\eta_2}),$$

where $\eta = \min\{r_0, (n + 1)\alpha - 1\}$ and $\eta_2 = (m + 1)\alpha - 1$.

*Proof.* Since $\Phi$ is of order $n$, it follows from (Redmann & Riedel, 2020, Theorem 4.2) that $\mathcal{R}(\Phi)$ has a global error of order $\eta_1 = \min\{r_0, (n + 1)\alpha - 1\}$. The global rate for the recovery of the initial condition $y_0$ from $\bar{y}_i$ follows immediately from Redmann & Riedel (2020, Proposition 4.1). $\square$

## 3.4 STABILITY OF EES SCHEMES FOR SDES

As discussed in the introduction and subsequently in Section 3, EES schemes offer a stable alternative to reversible integration. Whilst the stability of EES in the case of ODEs has been studied in Shmelev et al. (2025) and Section 3, we are interested in the stability of $\text{EES}_{\mathcal{R}}$ when applied to stochastic drivers for our applications to Neural SDEs. To evaluate the stability in the context of SDEs, we consider the *mean-square stability*, which has been widely used for the analysis of stochastic integration methods in the literature (Higham, 2000; Drummond & Mortimer, 1991; Hernandez & Spigler, 1993; Komori & Mitsui, 1995; Komori et al., 1994; Petersen, 1998; Saito & Mitsui, 1993; 1996; Schurz, 1996). Given the test equations

$$dy_t = \lambda y_t dt + \mu y_t dW_t, \tag{8}$$

where $\lambda, \mu \in \mathbb{C}$ and $y_0 \neq 0$ almost surely, a solution $\{y_n\}_{n \geq 0}$ derived from a numerical integrator is said to be *mean-square stable* if $\lim_{n \to \infty} \mathbb{E}(|y_n|^2) = 0$. It follows in a similar fashion to Theorem 3.1 that $\text{EES}(2, 5; x)$ applied to 8 is mean-square stable if and only if

$$\mathbb{E}\left[\left|1 + \rho + \frac{1}{2}\rho^2 + \frac{1}{8}\rho^3\right|^2\right] < 1,$$

where $\rho = \lambda dt + \mu dW_t \sim N(\lambda dt, \mu^2 dt)$. Figure 2 shows 4 cross-sections of the stability domain. For comparison, we take the RDE analogues of RK3 and RK4 of the form in equation 6, $\mathcal{R}(\text{RK3})$ and $\mathcal{R}(\text{RK4})$. Along most cross-sections, $\text{EES}_{\mathcal{R}}(2, 5)$ achieves a similar or greater stability to $\mathcal{R}(\text{RK3})$ and $\mathcal{R}(\text{RK4})$.

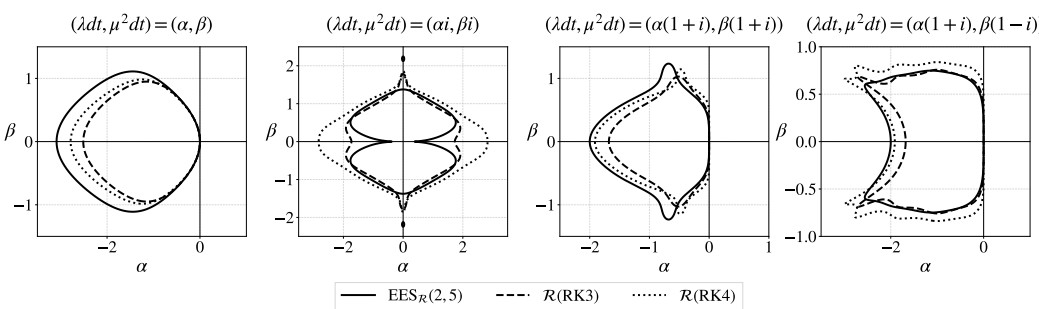

Figure 2: *Cross sections of the mean-square stability domains of* $\mathrm{EES}_{\mathcal{R}}(2,5)$, $\mathcal{R}(RK3)$ *and* $\mathcal{R}(RK4)$.

### 3.5 BACKPROPAGATION THROUGH EXPLICIT RUNGE–KUTTA METHODS

The algorithm for backpropagation through an explicit Runge–Kutta scheme $\Upsilon$ of the form in equation 6 is given in Algorithm 1. We assume the solver is applied to a (neural) RDE of the form

$$dy_t^h = f(y_t^h; \theta)d\mathbf{X}_t^h, \tag{9}$$

where $\theta$ are learnable parameters requiring backpropagation, trained with respect to a loss $L(\{y_n^h\}_{n=0}^N)$. As with all reversible schemes, a reverse step $\bar{\Upsilon}$ is used to recover $y_n$ from $y_{n+1}$, followed by a backpropagation through the internal operations of the solver $\Upsilon$. The latter step is achieved by defining $z_i = f(k_i; \theta)$ and computing the derivatives $\partial L/\partial z_i$ and $\partial L/\partial k_i$ in reverse through the stages $i = s, s-1, \ldots, 1$. At each stage, a backpropagation algorithm is called to backpropagate the derivative $\partial L/\partial z_i$ through $f$, resulting in the derivative $\partial L/\partial k_i$ and a local derivative with respect to $\theta$, $d_\theta$.

---

**Algorithm 1** Backpropagation through Explicit Runge–Kutta Schemes

**Input:** $y_{n+1}, \partial_{y_{n+1}}L$
**Input:** Running derivative with respect to $\theta$, $\partial_\theta L$
**Input:** Explicit RK method $\Upsilon$ of the form in equation 6 with coefficients $\{a_{ij}\}_{1 \leq i,j \leq s}$ and $\{b_i\}_{1 \leq i \leq s}$.

$\quad y_n = \bar{\Upsilon}(y_{n+1}, dX)$
$\quad$**for** $i = s, \ldots 1$ **do**
$\quad\quad \partial_{z_i}L = b_i dX \cdot \partial_{y_{n+1}}L + \sum_{j=i+1}^s a_{ji} dX \cdot \partial_{k_j}L$
$\quad\quad d_\theta, \partial_{k_i}L = \mathrm{backprop}_f(\partial_{z_i}L)$
$\quad\quad \partial_\theta L \mathrel{+}= d_\theta$
$\quad$**end for**
$\quad \partial_{y_n}L = \partial_{y_{n+1}}L + \sum_{i=1}^s \partial_{k_i}L$
$\quad$**return** $y_n, \partial_{y_n}L, \partial_\theta L$

---

## 4 EXPERIMENTS

We evaluate the performance of $\mathrm{EES}_{\mathcal{R}}(2,5)$ on examples of Neural SDEs with challenging dynamics. We note that there are limited solvers which can be used as baselines in our experiments, with Reversible Heun being the only widely adopted explicit reversible SDE solver at the time of writing. In order to expand our baselines, we take the approach of McCallum-Foster for constructing reversible ODE solvers and apply it to the RDE versions of the Euler and Explicit Midpoint schemes, defined by the form in equation 6. Algorithm 1 can then be used in conjunction with the backpropagation algorithm found in McCallum & Foster (2024) to backpropagate efficiently through the resulting schemes.

We present the results of training Neural SDEs on OU and GBM dynamics. In both cases, we pick the step sizes of the solvers to fix the total number of function evaluations of the drift and diffusion, leading to comparable runtimes for all of the solvers.

## 4.1 HIGH VOLATILITY ORNSTEIN–UHLENBECK PROCESS

Consider learning the Ornstein–Uhlenbeck (OU) dynamics

$$dy_t = \nu(\mu - y_t)dt + \sigma dW_t, \quad y_0 \in \mathbb{R},$$

under a high-volatility regime $\sigma \gg 0$. Specifically, we take $\nu = 0.2, \mu = 0.1$ and $\sigma = 2$. Motivated by Oh et al. (2024), we take a Neural Langevin SDE (LSDE) defined by

$$dz_t = g(z_t; \theta_g)dt + f(t; \theta_f) \circ dW_t, \quad z_0 = h(\mathbf{x}, \theta_h) \in \mathbb{R}^{d_z},$$

where $h$ is a learnable affine function of the input data $\mathbf{x} = \{x_n\}_{n \geq 0}, x_n \in \mathbb{R}^2$, sampled from the true OU dynamics, and $g, f$ are neural networks parametrised by $\theta_g, \theta_f$ respectively. We choose the dimension of the latent representation $d_z = 32$, and parametrise $f, g$ as 2-layer neural networks of width 32 with LipSwish activations. The SDEs are integrated over $t \in [0, 10]$ and the LSDE is trained for 250 epochs using the Adam optimiser with a fixed learning rate of $10^{-3}$. At each epoch, 50,000 realisations of the trained dynamics are sampled and the MSE loss is computed against the true OU dynamics.

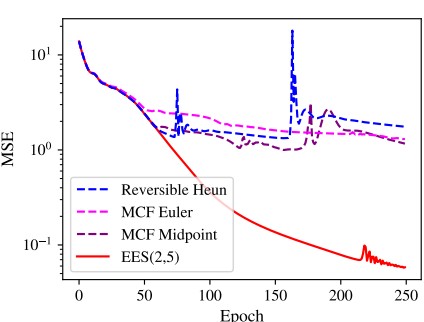

Figure 3: *Training MSE for OU dynamics with a fixed number of evaluations of $f, g$.*

Figure 3 shows the training loss using Reversible Heun, McCallum-Foster (MCF) methods and $\text{EES}_{\mathcal{R}}(2, 5)$, with the step size chosen such that the number of evaluations of $f, g$ is fixed between solvers. Such a choice results in comparable runtimes for all of the solvers, allowing for a fair comparison. Table 1 gives the number of evaluations of $f, g$ per step of the solvers, the chosen step size, the terminal MSE and the total runtime of each solver. From Figure 3, we see that for the initial $\sim 50$ epochs, the methods perform similarly. After this, $\text{EES}_{\mathcal{R}}(2, 5)$ significantly outperforms the other methods, suggesting the model has begun to learn high-volatility dynamics which cause instability in the Reversible Heun and McCallum-Foster methods.

| Method | #Eval. / Step | Step Size | Terminal MSE | Runtime (s) |
|---|---|---|---|---|
| Reversible Heun | 1 | 1/12 | 1.0190 | 368.2 |
| MCF Euler | 2 | 1/6 | 1.3048 | 307.5 |
| MCF Midpoint | 4 | 1/3 | 1.1651 | 279.5 |
| $\text{EES}_{\mathcal{R}}(2, 5)$ | 3 | 1/4 | 0.0582 | 261.3 |

Table 1: Metrics for OU dynamics. The step size is chosen such that the total number of evaluations of $f, g$ per integration is fixed.

## 4.2 HIGH-DIMENSIONAL GBM WITH STIFF DRIFT

Consider learning the dynamics of a high-dimensional geometric Brownian motion (GBM)

$$dy_t = Ay_t dt + \sigma y_t dW_t, \quad y_0 \in \mathbb{R}^d,$$

where $A \in \mathbb{R}^{d \times d}$ and $\sigma \in \mathbb{R}$. We introduce a stiff drift component by choosing $A = QDQ^T$, where $D = \text{diag}(\lambda_0, \lambda_1, \dots, \lambda_{d-1})$, $\lambda_i = -20(1 + \frac{i}{d})$, and $Q$ is a randomly generated orthogonal matrix, and take $d = 25$ and $\sigma = 0.1$. We choose to learn the dynamics using a Neural SDE of the form

$$dz_t = g(z_t; \theta_g)dt + f(z_t; \theta_f) \circ dW_t, \quad z_0 = h(\mathbf{x}, \theta_h) \in \mathbb{R}^{d_z},$$

where $f, g$ are neural networks with the same architecture as Section 4.1. We integrate the SDEs over $t \in [0, 1]$ for 1,000 epochs, sampling 10,000 realisations of the dynamics at every epoch. The Adam optimiser is used with a fixed learning rate of $2 \times 10^{-1}$.

As in the previous example, Figure 4 and Table 2 show the results of training using various reversible methods, with the step size chosen such that the number of evaluations of $f, g$ is fixed. We see that the

instability caused by the stiff drift results in diverging MSE for all solvers except $\text{EES}_{\mathcal{R}}(2,5)$, which manages to retain moderate stability for the entire 1,000 epochs of training. Figure 5 shows the MSE of the gradient of the loss during training, where the true gradient is computed by autodifferentiation through a discretise-then-optimise solution, using the same solver and step size. Despite its near-reversibility, $\text{EES}_{\mathcal{R}}(2,5)$ achieves a lower gradient MSE compared to other solvers. This effect is likely the result of the superior stability of $\text{EES}_{\mathcal{R}}(2,5)$, in combination with the linear nature of the target SDE.

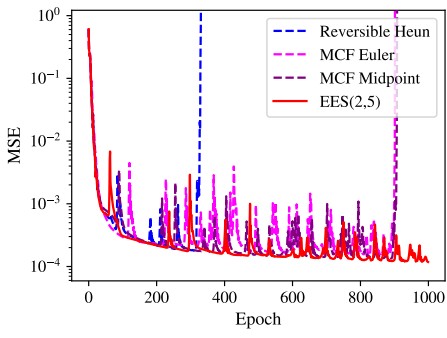

Figure 4: *Training MSE for GBM dynamics with a fixed number of evaluations of $f, g$.*

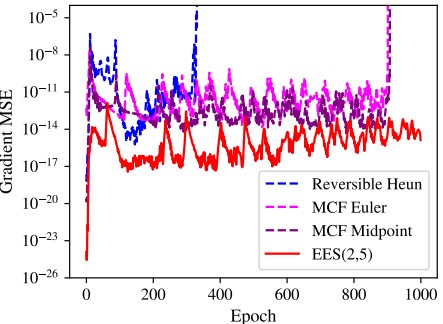

Figure 5: *Gradient MSE for GBM dynamics with a fixed number of evaluations of $f, g$.*

| Method | #Eval. / Step | Step Size | Terminal MSE | Runtime (s) |
|---|---|---|---|---|
| Reversible Heun | 1 | 1/60 | - | 1283.6 |
| MCF Euler | 2 | 1/30 | - | 1119.9 |
| MCF Midpoint | 4 | 1/15 | - | 1270.1 |
| $\text{EES}_{\mathcal{R}}(2,5)$ | 3 | 1/20 | 1.1803E-4 | 1050.0 |

Table 2: Metrics for stiff GBM dynamics. The step size is chosen such that the total number of evaluations of $f, g$ per integration is fixed.

## 5  CONCLUSIONS, LIMITATIONS AND FUTURE WORK

In this paper, we have introduced the use of Explicit and Effectively Symmetric (EES) schemes as stable integrators for Neural SDEs, and more generally, RDEs. Using the framework of Redmann & Riedel (2020), we have adapted existing EES schemes introduced by Shmelev et al. (2025) for ODEs to the more general setting of RDEs. Through *mean-square stability* analysis, we have shown that EES schemes possess similar stability domains to classical RK3 and RK4 schemes when applied to SDEs. We discussed an efficient algorithm for backpropagation through explicit Runge–Kutta schemes for RDEs, and presented two experiments involving the training of Neural ODEs on extreme SDE dynamics. In both experiments, our $\text{EES}_{\mathcal{R}}(2,5)$ scheme demonstrated superior stability to the Reversible Heun scheme, resulting in faster training and a lower terminal loss.

EES schemes provide a stable alternative to existing methods such as Reversible Heun, resulting in fast and accurate training when applied to complex dynamics. However, when stability is not an issue, Reversible Heun offers faster integration by requiring only one evaluation of the drift and diffusion per step, as opposed to 3 evaluations required by $\text{EES}_{\mathcal{R}}$. Addressing this limitation, either through algorithmic changes or the derivation of new EES-type schemes which employ auxiliary variables, is left for future work.

There are several potential extensions to this paper which are left for future research. Applications of EES schemes to more complicated models, including but not limited to Neural Jump SDEs (Jia & Benson, 2019; Herrera et al., 2020), Neural CDEs (Kidger et al., 2020) and Neural RDEs (Morrill et al., 2021), may be of interest. An extension of EES schemes to include partitioned or adaptive step-size schemes would prove valuable for the training of stiff neural differential equations.

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

# A ROOTED TREES AND B-SERIES

## A.1 THE CONNES-KREIMER HOPF ALGEBRA

We give a brief account of non-planar (labelled) rooted trees and the Connes-Kreimer Hopf algebra. We refer the reader to Hoffman (2003) for a comprehensive presentation. A non-planar labelled rooted tree is defined as a graph $\tau = (V, E, r)$ with vertex set $V$, edge set $E$ and a root vertex $r \in V$, together with a set of vertex decorations drawn from $\{1, \ldots, d\}$. We denote the empty tree by $\emptyset$. Given trees $\tau_1, \ldots, \tau_m$, we write $[\tau_1, \ldots, \tau_m]_a$ to denote the tree formed by connecting the root vertices of $\tau_1, \ldots, \tau_m$ to a new root, which receives the label $a \in \{1, \ldots, d\}$. Non-planarity means that the order of the trees in $[\tau_1, \ldots, \tau_m]_a$ is irrelevant. Repeated trees will be denoted using power notation, for instance

$$[\tau_1, \tau_1, \tau_2, \tau_3, \tau_3, \tau_3]_a = [\tau_1^2, \tau_2, \tau_3^3]_a.$$

We write $|\tau|$ to denote the number of vertices in a tree. Additionally, we define the following combinatorial quantities, defined on unlabelled trees:

$$\emptyset! = 1, \quad \bullet! = 1, \quad [\tau_1, \ldots, \tau_m]! = |[\tau_1, \ldots, \tau_m]| \prod_{i=1}^{m} \tau_i!,$$

$$\sigma(\emptyset) = 1, \quad \sigma(\bullet) = 1, \quad \sigma([\tau_1^{k_1}, \ldots, \tau_m^{k_m}]) = \prod_{i=1}^{m} k_i! \sigma(\tau_i)^{k_i},$$

$$\beta(\emptyset) = 1, \quad \beta(\bullet) = 1, \quad \beta([\tau_1^{k_1}, \ldots, \tau_m^{k_m}]) = \binom{[\tau_1^{k_1}, \ldots, \tau_m^{k_m}]}{|\tau_1|, \ldots, |\tau_m|} \prod_{i=1}^{m} \frac{1}{k_i!} \beta(\tau_i)^{k_i}.$$

We will refer to the commutative juxtaposition of trees as a forest. We write $\mathcal{T}$ to denote the set of all non-planar labelled rooted trees, and $\mathcal{T}_N \subset \mathcal{T}$ to denote the trees $\tau$ with $|\tau| \leq N$. The free commutative $\mathbb{R}$-algebra generated by $\mathcal{T}$ will be denoted $\mathcal{H}$. The Connes-Kreimer (Connes & Kreimer, 1999) Hopf algebra on $\mathcal{H}$ is defined as follows. Multiplication $\mu : \mathcal{H} \otimes \mathcal{H} \to \mathcal{H}$ is defined as the commutative juxtaposition of two forests, extended linearly to $\mathcal{H}$. The multiplicative unit is defined to be the empty forest $\emptyset$. The counit map $\varepsilon : \mathcal{H} \to \mathbb{R}$ is defined by $\varepsilon(\emptyset) = 1$ and $\varepsilon(\tau) = 0$ for all non-empty trees $\tau \in \mathcal{H}$. The coproduct map is defined recursively by

$$\Delta(\emptyset) = \emptyset \otimes \emptyset,$$
$$\Delta[\tau_1, \ldots, \tau_m]_a = [\tau_1, \ldots, \tau_m]_a \otimes \emptyset + (\mathrm{id} \otimes B_+^a)(\Delta \tau_1 \cdots \Delta \tau_m),$$

where $B_+^a(\tau_1 \cdots \tau_m) := [\tau_1 \cdots \tau_m]_a$ for a forest $\tau_1 \cdots \tau_m$. The definition is extended to a linear multiplicative map on $\mathcal{H}$. We will occasionally use Sweedler's notation

$$\Delta \tau = \sum_{(\tau)} \tau^{(1)} \otimes \tau^{(2)}$$

for the coproduct. We omit the definition of the antipode $S$ here, and instead refer the reader to (Manchon, 2004; Hoffman, 2003). We denote the dual of the Connes-Kreimer Hopf algebra by $\mathcal{H}^*$. For $\varphi_1, \varphi_2 \in \mathcal{H}^*$, the convolution product is defined by

$$\varphi_1 * \varphi_2 = \mu_{\mathbb{R}} \circ (\varphi_1 \otimes \varphi_2) \circ \Delta,$$

with $\mu_{\mathbb{R}} : \mathbb{R} \otimes \mathbb{R} \to \mathbb{R}$ denoting multiplication in $\mathbb{R}$.

## A.2 B-SERIES EXPANSIONS OF ODES

For any tree $\tau \in \mathcal{T}$, the so-called elementary differential $F(\tau)(y)$ (Butcher, 2016) is defined recursively by

$$F(\emptyset)(y) = y, \quad F(\bullet_i)(y) = f_i(y),$$
$$F([\tau_1, \tau_2, \ldots, \tau_m]_i)(y) = f_i^{(m)}(y)(F(\tau_1)(y), F(\tau_2)(y), \ldots, F(\tau_m)(y)).$$

Given a map $\varphi : \mathcal{T} \to \mathbb{R}$, the associated B-series is defined

$$B_h(\varphi, y_0) := \sum_{\tau \in \mathcal{T}} \frac{h^{|\tau|}}{\sigma(\tau)} \varphi(\tau) F(\tau)(y_0).$$

A key property of B-series is their closure under composition. One can show that for two B-series Hairer & Wanner (1974); Butcher et al. (2024),

$$B_h(\varphi_2, B_h(\varphi_1, y_0)) = B_h(\varphi_1 * \varphi_2, y_0),$$

where $\varphi_1 * \varphi_2$ is the convolution product defined above. It can be shown that the exact solution to the ODE in equation 2 admits a B-series representation $y(h) = B_h(e, y_0)$, where $e(\tau) = 1/\tau!$. Similarly, the solution given by a Runge–Kutta scheme with coefficients $\{a_{ij}\}_{1 \le i,j \le s}$, $\{b_i\}_{1 \le i \le s}$ admits the B-series representation $B_h(\varphi, y_0)$, where (Butcher, 2016, Lemma 312B)

$$\varphi(\tau) := \sum_{i_1,\dots,i_n} b_{i_1} \prod_{(k,\ell) \in E} a_{i_k,i_\ell}$$

for a tree $\tau = (V, E, r)$ with $|\tau| = n$.

We refer the reader to (Hairer et al., 2006; McLachlan et al., 2015; Butcher, 2021) for a detailed account of B-series and the Butcher group.

## A.3 BRANCHED ROUGH PATHS

Let $\mathcal{H}$ be the Connes-Kreimer Hopf algebra of non-planar labelled rooted trees defined above.

**Definition A.1.** Let $\alpha \in (0, 1]$. An $\alpha$-Hölder branched rough path is a map $\mathbf{X} : [0, T]^2 \to \mathcal{H}^*$ such that

1. for all $s, t \in [0, T]$ and $\tau_1, \tau_2 \in \mathcal{H}$,
$$\langle \mathbf{X}_{s,t}, \tau_1 \rangle \langle \mathbf{X}_{s,t}, \tau_2 \rangle = \langle \mathbf{X}_{s,t}, \tau_1 \tau_2 \rangle,$$

2. for all $\tau \in \mathcal{H}$,
$$\langle \mathbf{X}_{s,t}, \tau \rangle = \sum_{(\tau)} \langle \mathbf{X}_{s,u}, \tau^{(1)} \rangle \langle \mathbf{X}_{u,t}, \tau^{(2)} \rangle,$$

   where $\Delta \tau = \sum_{(\tau)} \tau^{(1)} \otimes \tau^{(2)}$.

3. for all $\tau \in \mathcal{H}$,
$$\sup_{s \ne t} \frac{|\langle \mathbf{X}_{s,t}, \tau \rangle|}{|t - s|^{\alpha|\tau|}} < \infty.$$

*Remark.* As remarked in (Hairer & Kelly, 2015; Gubinelli, 2010), the components $\langle \mathbf{X}_{s,t}, \tau \rangle$ with $|\tau| > N$ are determined by those with $|\tau| \le N$, where $N$ is the largest integer such that $N\alpha \le 1$.

The space of $\alpha$-Hölder branched rough paths is a complete metric space under the metric

$$\varrho_\alpha(\mathbf{X}, \mathbf{Y}) := \sum_{\tau \in \mathcal{T}_N} \sup_{s \ne t} \frac{|\langle \mathbf{X}_{s,t} - \mathbf{Y}_{s,t}, \tau \rangle|}{|t - s|^{\alpha|\tau|}},$$

where $N = \lfloor 1/\alpha \rfloor$.

## A.4 B-SERIES EXPANSIONS OF RDES

We recount the results of (Redmann & Riedel, 2020) regarding series expansions of the solutions to RDEs of the form equation 4. Recall the definition of the elementary differential $F(\tau)(y)$ from Appendix A.2.

*Theorem A.1* ((Redmann & Riedel, 2020, Theorem 2.10)). Let $\mathbf{X}$ be an $\alpha$-branched rough path and $h > 0$. Then equation 4 admits the series expansion

$$y(t_0 + h) = y_0 + \sum_{\tau \in \mathcal{T}_p} \frac{1}{\sigma(\tau)} F(\tau)(y_0) \langle \mathbf{X}_{t_0,t_0+h}, \tau \rangle + \mathcal{O}(h^{(p+1)\alpha})$$

for all $p \ge \lfloor 1/\alpha \rfloor$.

## B   GENERAL RUNGE–KUTTA METHODS FOR RDEs

Following Burrage & Burrage (1996; 1998; 2000); Redmann & Riedel (2020), consider the class of general Runge-Kutta methods defined by

$$y_{n+1} = y_n + \sum_{m=1}^{d} \sum_{i=1}^{s} z_i^{(m)} f_m(k_i), \tag{10}$$

$$k_i = y_n + \sum_{m=1}^{d} \sum_{j=1}^{s} Z_{ij}^{(m)} f_m(k_j),$$

where $Z^{(1)}, \ldots, Z^{(d)} \in \mathbb{R}^{s \times s}$ and $z^{(1)}, \ldots, z^{(d)} \in \mathbb{R}^s$. We briefly recount the local and global error rates for such schemes, as formulated in Redmann & Riedel (2020). The results are based on the adaptation of B-series to RDEs presented above.

**Definition B.1.** Given $h > 0$, define the maps $a, \varphi$ recursively over non-planar labelled rooted trees $\tau$ by setting $\varphi(\emptyset)(h) := (1, \ldots, 1)^T \in \mathbb{R}^s$, where $\emptyset$ denotes the empty tree, and for a tree $\tau = [\tau_1 \cdots \tau_n]_i$ formed by joining $\tau_1, \ldots, \tau_n$ by a new root labelled $i$,

$$\varphi(\tau)(h) := \prod_{j=1}^{n} (Z^{(i)} \varphi(\tau_j)(h)),$$

$$a(\tau)(h) := \left\langle z^{(i)}, \prod_{j=1}^{n} \varphi(\tau_j)(h) \right\rangle.$$

*Theorem* B.1 ( (Redmann & Riedel, 2020)). The general Runge-Kutta method given by equation 10 has a local error of order $(p+1)\alpha$ if and only if

$$\langle \mathbf{X}_{t_0, t_0+h}, \tau \rangle = a(\tau)(h)$$

for all non-planar labelled rooted trees $\tau$ with $p$ or fewer nodes, i.e. $|\tau| \leq p$.

*Proposition* B.1 ((Redmann & Riedel, 2020, Proposition 4.1)). Let $y(t, y_0)$ denote the solution to equation 4 at time $t$ starting at $y_0$. Suppose the Runge–Kutta method equation 10 has a local error of order $(p+1)\alpha$, and there exists a constant $C_1 > 0$ such that

$$|y(h, y_0) - y(h, \widetilde{y}_0)| \leq C_1 |y_0 - \widetilde{y}_0|,$$

for $h$ sufficiently small. Then there exists $C > 0$ such that

$$\max_{n=0,\ldots,N} |y(t_n) - y_n| \leq C h^{(p+1)\alpha - 1}.$$

### B.1   CONVERGENCE OF SIMPLIFIED RUNGE–KUTTA METHODS

*Theorem* B.2 ((Redmann & Riedel, 2020, Theorem 3.3)). Let $\Phi$ be an ODE Runge–Kutta scheme. The Runge-Kutta method $\mathcal{R}(\Phi)$ approximating $y^h$ has a local error of order $(p+1)\alpha$, i.e.

$$y^h(t_0 + h) - y_1^h = \mathcal{O}(h^{(p+1)\alpha}),$$

if and only if the ODE Runge–Kutta method $\Phi$ is of order $p$.

*Theorem* B.3 ((Redmann & Riedel, 2020, Theorem 4.2)). Let $\Phi$ be an ODE Runge–Kutta method of order $p$. Suppose that $f$ is $\mathrm{Lip}_b^\gamma$ for some $\gamma > 1/\alpha$. Then $\mathcal{R}(\Phi)$ has a global error rate of $\eta = \min\{r_0, (p+1)\alpha - 1\}$, where $r_0$ is the convergence rate of the Wong-Zakai approximation. That is,

$$\max_{n=0,\ldots N} |y(t_n) - y_n^h| = \mathcal{O}(h^\eta).$$

## C   CONVERGENCE OF $\mathrm{EES}_\mathcal{R}$ SCHEMES

We verify the global error rates given in Theorem 3.3 experimentally by reproducing the example given in (Redmann & Riedel, 2020; Deya et al., 2012) for $\mathrm{EES}_\mathcal{R}(2, 5; 1/10)$ and $\mathrm{EES}_\mathcal{R}(2, 5; (5 - 3\sqrt{2})/14)$. We take the RDE

$$dy_t = \cos(y_t) d\mathbf{X}_t^{(1)} + \sin(y_t) d\mathbf{X}_t^{(2)}, \quad y_0 = 1$$

for $t \in [0, 1]$, where $\mathbf{X}$ is the geometric lift pf a 2-dimensional fractional Brownian motion (fBm) with hurst index $H$. We compute the average of the maximal discretization error over $M = 10$ realisations of the RDE,

$$\mathcal{E}(h) := \frac{1}{M} \sum_{i=1}^{M} \max_{n=0,\dots,N} |y_i(t_n) - y_{i,n}|,$$

where $y_i(t)$ denotes the solution to the $i^{th}$ realisation of the RDE and $y_{i,n}$ denotes the discretisation of the $i^{th}$ solution using an $\text{EES}_{\mathcal{R}}$ scheme. Additionally, we evaluate the average error when recovering the initial condition,

$$\bar{\mathcal{E}}(h) := \frac{1}{M} \sum_{i=1}^{M} |y_0 - \bar{\bar{y}}_{i,n}|.$$

From (Friz & Riedel, 2014), the rate $r_0$ can be chosen arbitrarily close to $2H - 1/2$ for a fractional Brownian motion with Hurst parameter $H$. It follows that we expect $\eta_1 = 2H - 1/2$ in Theorem 3.3 for both $\text{EES}(2, 5)$ and $\text{EES}(2, 7)$, and $\eta_2 = 6H - 1$ for $\text{EES}(2, 5)$ and $\eta_2 = 8H - 1$ for $\text{EES}(2, 7)$. These rates are shown in Figures 6, 7 and 8 for $H = 0.4, 0.5$ and $0.6$ respectively.

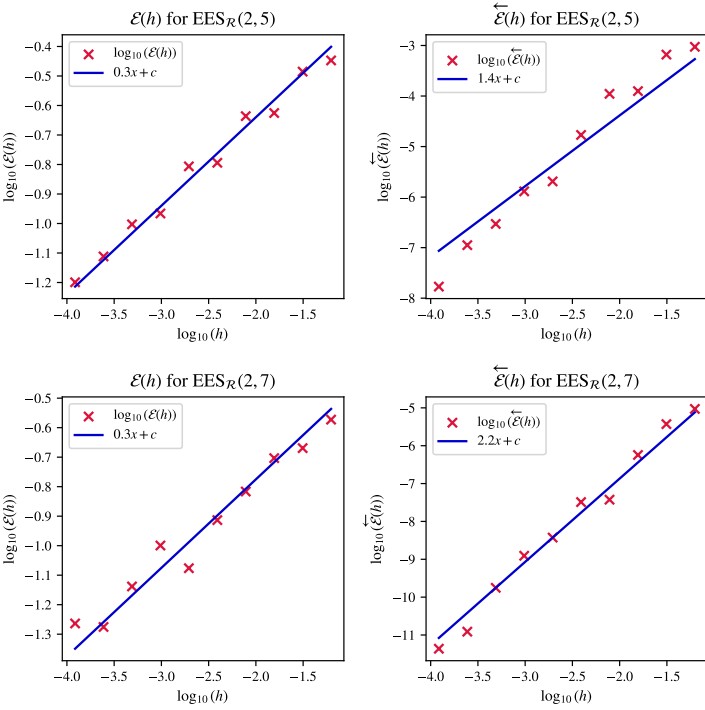

Figure 6: *Convergence rates for $H = 0.4$*

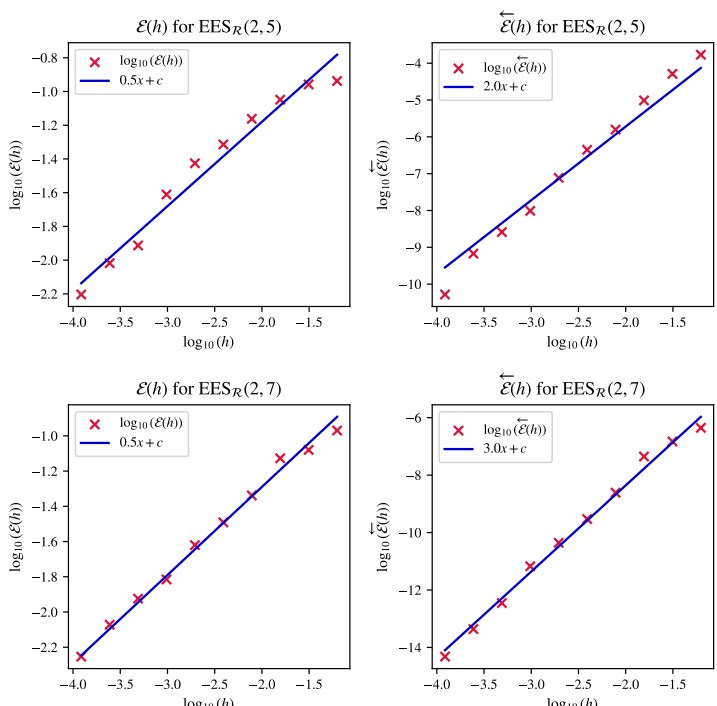

Figure 7: *Convergence rates for $H = 0.5$*

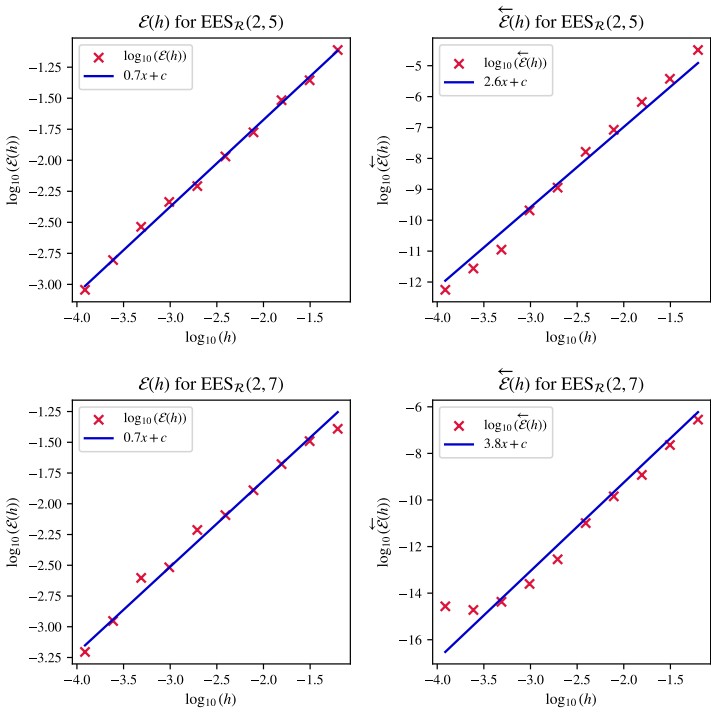

Figure 8: *Convergence rates for $H = 0.6$*