# OpenReview forum: "Explicit and Effectively Symmetric Schemes for Neural SDEs"
_ICLR.cc/2026/Conference — Submitted to ICLR 2026_

### Official Review · Reviewer_VyCj · 2025-10-28

**Soundness:** 4
**Presentation:** 4
**Contribution:** 2
**Rating:** 4
**Confidence:** 3

**Summary:**

Constructing reversible solvers for SDEs is an important problem in Neural SDEs due to computational problems with both optimize-then-discretize and discretize-then-optimize approaches. Existing reversible solvers such as ALF and Reversible Heun have a small stability region, making them unsuitable for many applications. The author construct an RDE version of the EES scheme, a type of RK scheme, and show that it has a comparable stability region as well-established RK schemes. Numerical experiments are provided where EES_R outperforms Reversible Heun.

**Strengths:**

- Good writing with a good level of rigor. The introduction of the related works makes the method a very natural continuation.
- EES_R seems to inherit all the good properties of EES.
- EES_R has good performance on two test equations.

**Weaknesses:**

- Lack of novel contribution: I feel that EES_R is an application of the result from Redmann & Riedel to EES.
- The experiment with SDE-GAN from [1] is missing, which makes it a bit hard to fully appreciate that EES_R is a strict improvement over Rev Heun.
[1] Kidger, Patrick, et al. "Efficient and accurate gradients for neural sdes." Advances in Neural Information Processing Systems 34 (2021): 18747-18761.

**Questions:**

- In Figure 1, is it possible to plot the stability region of Reversible Heun? It is unclear to me what the metric to compare the stability region of Rev Heun vs EES_R.
- I'm guessing that one (albeit very small) drawback of EES_R vs Reversible Heun is simply the extra computation/memory to evaluate the drift and diffusion term. Maybe the difference is only a constant depending on the order of EES. Is that true?
- It is not clear to me what the domains in the two experiments are and how much they overlap with the stability region of EES and Reversible Heun.
- I would like to understand more about the near-symmetry property. It is fine if near-symmetry is enough for practical problems, but is it possible to construct pathological test equations such that near-symmetry is not enough to guarantee the accuracy of the inverse solution is bounded? What are the properties of these pathological cases such that they differ from the typical average-case practical problems?
- Can you be more precise about the "high-volatility dynamics" mentioned on line 392?

---

> ### Author Response · Authors · 2025-11-20
>
> We thank the reviewer for their time and helpful feedback. We hope that the points below address all the reviewer's concerns.
>
> **Weaknesses:**
>
> Regarding the concern about lack of novelty, we would like to clarify that, to the best of our knowledge, our method is the first explicit reversible SDE solver with a large stability region. This constitutes a substantive theoretical advance, as existing reversible schemes are either implicit or possess significantly more restrictive stability regions. In addition, while Redmann & Riedel introduce a general class of RK schemes for RDEs, they do not address the question of reversibility. Our contribution is to identify and formalise the precise structural conditions under which a RK scheme becomes reversible and can be made explicit in the SDE setting—something that is neither contained in nor implied by their analysis, and represents, we believe, a genuinely new component of our work.
>
> We agree that the original baseline was narrow. As far as we are aware, Reversible Heun is the only available explicit reversible SDE solver. Other schemes such as ALF and McCallum-Foster are ODE-only methods. To expand the baselines, we have adapted the original ODE McCallum–Foster method to the SDE setting by applying their techniques to RDE Runge-Kutta schemes, and we include two variants of this solver (using Euler and Midpoint discretisations) across all experiments. Additionally, we have now replaced the original option price GBM example with an example of high-dimensional GBM dynamics with stiff drift, where the drift is given by A y_t dt for a matrix A with extreme negative eigenvalues. We believe this example is a clearer demonstration of the failure modes of existing reversible solvers. Please see Section 4.2 in the revised paper.
>
> To deepen the analysis of our experiments, we have added the MSE of the gradients of the loss during training for the second example, computed against reference gradients of a discretise-then-optimise solution. Please see Section 4.2 and Figure 5.
>
> **Questions:**
>
> * The stability region of Reversible Heun is the interval [-i, i], so it can't be plotted as it has no area. We have added a note on line 235.
> * Yes, EES does require additional drift and diffusion evaluations per step, and this cost grows with the order of the scheme. In the revised paper, we address this by selecting each solver’s step size so that all methods perform the same number of drift and diffusion evaluations. This approximately equalises runtime and enables a fair comparison. Under this fixed computational budget, it is now clear that EES consistently outperforms the other reversible methods.
> * The stability domains we present in the context of theoretical mean-square stability analysis show the convergence of solutions on the linear test problem $dy_t = \lambda y_t dt + \mu y_t dW_t$ as $\lambda h$ and $\mu h$ change, where h is the step size. As such, stability domains refer to behaviour over all possible linear problems, and do not make claims about general non-linear problems. As such, the experiments themselves do not possess a “domain”.
> * Yes, it is indeed possible to construct pathological examples where near-symmetry is insufficient. Since EES(2,5) reconstructs the inverse flow up to order 5, one can design vector fields whose fifth derivative is unbounded. In such cases, near-symmetry alone does not guarantee stability of the backward reconstruction. However, these pathologies are not specific to reversibility. For such badly behaved vector fields, the forward solution itself would already incur large discretisation errors with any low-order method, long before issues arise from the backward pass. In all the practical settings we considered in this paper, near-symmetry provides a highly accurate and robust approximation of the inverse flow, and we have not observed instability arising from backward reconstruction in any of our empirical tests.
> * The "high-volatility dynamics" refer to the choice of sigma = 2. As the Neural SDE learns, the learnt vector field approaches that of the high-volatility target SDE, making the dynamics hard to integrate.  As shown in Figure 3, this causes instability in existing solvers after epoch ~50, but does not affect the EES scheme, which has a higher stability.

---

> > ### Comment · Reviewer_VyCj · 2025-11-24
> >
> > Thank you the authors for the clarifications! I have raised my score.

---

### Official Review · Reviewer_tYYW · 2025-10-28

**Soundness:** 3
**Presentation:** 3
**Contribution:** 3
**Rating:** 6
**Confidence:** 3

**Summary:**

The paper “Explicit and Effectively Symmetric Schemes for Neural SDEs” introduces a new family of Explicit and Effectively Symmetric (EES) Runge–Kutta solvers for Neural Stochastic Differential Equations (Neural SDEs). These methods aim to combine the stability and memory efficiency of reversible solvers with the simplicity and computational efficiency of explicit schemes.

Building upon recent advances in rough path theory, the authors generalize EES methods (originally designed for ODEs) to rough differential equations (RDEs). They derive theoretical results showing that EES schemes maintain high stability comparable to classical RK3/RK4 solvers and demonstrate mean-square stability for stochastic systems. Two experiments (a high-volatility Ornstein–Uhlenbeck process and a geometric Brownian motion calibration) show that the proposed EESR(2,5) method is significantly more stable and achieves faster, more reliable training than the commonly used Reversible Heun scheme.

**Strengths:**

Novelty & Contribution – Extends the concept of near-reversible explicit solvers (EES) to Neural SDEs/RDEs, bridging rough path theory and practical neural training.

Theoretical Rigor – Provides clear derivations of convergence and stability properties, including mean-square stability analysis.

Practical Relevance – Demonstrates tangible improvements in training stability on challenging stochastic models where existing reversible methods fail.

Clarity of Design – The algorithmic formulation for backpropagation (Algorithm 1) is well structured and easily integrable into autodiff frameworks.

Balanced Discussion – Acknowledges trade-offs (extra evaluations per step vs. stability) and outlines realistic future extensions.

**Weaknesses:**

Stability under Non-smooth Neural Coefficients:
The stability analysis assumes the drift and diffusion coefficients f,g are sufficiently smooth (bounded derivatives, Lipschitz).
→ How would the EESR scheme behave if the neural network coefficients are non-smooth or poorly regularized, as often happens in real-world Neural SDE training?
Would the mean-square stability guarantee still approximately hold, or could the method exhibit spurious oscillations?

Adaptive Step Size:
Could an adaptive step-size EESR be designed to dynamically maintain stability without sacrificing explicitness?

Computational Cost:
Given that EESR(2,5) requires three function evaluations per step (vs. one in Reversible Heun), how does this scale in large neural SDEs or diffusion models?

Stochastic Gradient Interaction:
When EESR is used inside a stochastic training loop (mini-batch + noisy gradients), is its near-reversibility still advantageous, or do stochastic gradient perturbations break that property?

Empirical Testing Scope:
The benchmarks focus on low-dimensional dynamics. How would EESR perform on high-dimensional latent Neural SDEs or in score-based diffusion models with neural drift and diffusion?

**Questions:**

See weakness

---

> ### Author Response · Authors · 2025-11-20
>
> We thank the reviewer for their time and helpful feedback. We hope that the points below address all the reviewer's concerns.
>
> **Weaknesses:**
>
> * Stability under Non-smooth Neural Coefficients: Theoretical stability analysis concerns the behaviour of solvers on linear SDEs. In some cases, non-linear SDEs can be analysed the same way via a linear eigenvalue decomposition, although stability analysis does not generally make theoretical guarantees about the non-linear case. In the revised paper, we have extended our experimental section to showcase instabilities which arise while training with existing reversible solvers and demonstrated the ability of EES to overcome these issues. For the second example, we have added the MSE of the gradients of the loss during training, computed against reference gradients of a discretise-then-optimise solution. Additionally, we have now replaced the original option price GBM example with an example of high-dimensional GBM dynamics with stiff drift, where the drift is given by A y_t dt for a matrix A with extreme negative eigenvalues. This example shows how the stability guarantees of EES hold when those of other solvers fail. Please see Sections 4.1 and 4.2 in the revised paper.
> * Adaptive Step Size: We have briefly considered the extension to adaptive step sizes, but left it for future work. Efficient adaptive step size RK schemes typically consist of two methods which share common intermediate steps. Due to the additional constraints imposed on the Butcher tableau by near-symmetry, it is not immediately clear how to construct such schemes.
> * Computational Cost: In the revised paper, we select each solver’s step size to fix the number of drift and diffusion evaluations. This approximately equalises runtime across solvers and enables a clearer comparison. Under this matched computational budget, EES still consistently outperforms all other reversible methods.
> * Stochastic Gradient Interaction: In stochastic training loops, the stochastic perturbations are applied globally to the gradients of the entire SDE solution. As such, they do not change the local gradient computation, and near-reversibility still maintains all of its benefits.
> * Empirical Testing Scope: We agree that the original set of baselines was limited. To the best of our knowledge, Reversible Heun is the only existing explicit reversible SDE solver. Other reversible schemes, such as ALF and McCallum–Foster, are designed for ODEs only. To broaden the comparison, we have adapted the ODE-based McCallum–Foster method to the SDE setting and included two variants (with Euler and Midpoint discretisations) as baselines across all experiments. Since ALF shares the same stability region as Reversible Heun in the ODE case, we have not included it. Regarding the dimensionality of our test problems, while the OU example is low-dimensional in its observation space, the underlying latent SDE is 32-dimensional, making this a genuinely high-dimensional setting for the solver. Nonetheless, we have replaced the original option-pricing GBM example with a high-dimensional GBM model featuring stiff drift, further expanding the empirical scope. Please see sections 4.1 and 4.2 in the revised paper.

---

> > ### Comment · Reviewer_tYYW · 2025-11-24
> >
> > I would like to thank the author(s) for the detailed reply. Most of my concerns have been addressed. But could the author(s) highlight the change with a different color in the revised manuscript? It is difficult to check if my concerned parts have been improved.

---

> > > ### Author Response · Authors · 2025-11-25
> > >
> > > Please see the latest version of the manuscript, where we have highlighted the revised text in red. Additionally, figures 3,4,5 and tables 1,2 are part of the revision.

---

### Official Review · Reviewer_G7my · 2025-11-01

**Soundness:** 4
**Presentation:** 3
**Contribution:** 2
**Rating:** 2
**Confidence:** 3

**Summary:**

The authors introduce explicit and effectively symmetric Runge-Kutta schemes to overcome the numerical stability issues of Reversible Heun methods.

**Strengths:**

The paper is very well written and provides a great introduction and background to the problem. Tackling stability issues in algebraically reversible solvers to overcome limitations in the other two alternative methods is is timely, and the construction is plausible and well-motivated.

**Weaknesses:**

Analysis is solely focused on comparisons between reversible Heun and EES(2,5). The scope of the theoretical contributions are limited to stability analysis of EES for stochastic SDE where analysis of ODEs has been shown in prior work. Without broader baselines or diffusion/practical Neural-SDE tasks, it’s hard to gauge practical impact.

**Questions:**

**Q1. Practical evidence of overcoming Reversible Heun failure modes.** The authors cite [1] in the context of the limitations of Reversible Heun, but do not provide empirical evidence that their EES(2,5) techniques have overcome these limitations. The work would greatly benefit from including a comparison on the applications in [1] and a comparison to the Euler integration used in [1].

[1] Qinsheng Zhang and Yongxin Chen. Path integral sampler: a stochastic control approach for sampling. arXiv preprint arXiv:2111.15141, 2021.

**Q2. Missing reversible/non-reversible baselines.** The work would greatly improve with comparisons to the ALF and the McCallum & Foster method.

**Q3. Relation to recent algebraically reversible RK solvers.** The authors should include a discussion of highly related recent work [2] specifically developing algebraically reversible explicit RK schemes, and should comment on the differences of their work.

[2] Zander W. Blasingame and Chen Liu, REX: Reversible Solvers for Diffusion Models. arXiv preprint arXiv:2502.08834v1, 2025

**Q4. Practical applications.** The work would greatly improve with numerical results comparing their technique to others on problems of high relevance to ICLR including diffusion/application based Neural-SDEs.

---

> ### Author Response · Authors · 2025-11-20
>
> We thank the reviewer for their time and helpful feedback. We hope that the points below address all the reviewer's concerns.
>
> **Weaknesses:**
>
> We agree that the original set of baselines was somewhat limited. To the best of our knowledge, Reversible Heun remains the only available explicit reversible SDE solver. Other reversible schemes, such as ALF and McCallum–Foster, are designed for ODEs only. To broaden the baselines, we have now adapted the original ODE McCallum–Foster method to the SDE setting by applying their techniques to RDE Runge-Kutta schemes, and we include two variants of this solver (using Euler and Midpoint discretisations) across all experiments. Since ALF shares the same stability region as Reversible Heun in the ODE setting, we have not included it among the baselines. In addition, we have standardised the computational budget across solvers by choosing step sizes so that each method performs the same number of drift and diffusion evaluations. This approximately equalises runtime and enables a clearer comparison of method performance. Please see Sections 4.1 and 4.2 in the revised paper.
>
> In the revised paper, we have now replaced the original option price GBM example with an example of high-dimensional GBM dynamics with stiff drift, where the drift is given by A y_t dt for a matrix A with extreme negative eigenvalues. We believe this example is a clearer demonstration of the failure modes of existing reversible solvers. Please see Section 4.2 in the revised paper.
> To deepen the analysis of our experiments, we have added the MSE of the gradients of the loss during training for the second example, computed against reference gradients of a discretise-then-optimise solution. Please see Section 4.2 of the revised paper.
>
> Regarding the concern about limited contributions, we would like to clarify that, to the best of our knowledge, our method is the first explicit reversible SDE solver with a large stability region. This represents a substantive theoretical advance, as existing reversible schemes are either implicit or have significantly more restrictive stability regions. Moreover, the additional experiments we have included in the revised manuscript highlight the advantages of our solver across a broader set of settings, further demonstrating the practical impact of this contribution.
>
> **Questions:**
>
> * Unfortunately, the results of [1] are difficult to reproduce, as the codebase is outdated and no longer maintained. In the revised paper, we have substantially expanded and strengthened our baseline comparisons to better illustrate the failure modes of Reversible Heun and McCallum-Foster methods, as we have described in the points above. We hope that these additions provide clear empirical evidence of the advantages offered by our EES schemes.
> * As mentioned above, we have now included McCallum-Foster methods in our baselines.
> * We note that reversible schemes for diffusions such as the REX schemes in [2] are adapted specifically for diffusion models and require a different form to general reversible solvers for SDEs. Whilst an adaptation of EES schemes to diffusions may be possible, this requires substantial changes which are outside the scope of the current paper.
> * As mentioned above, we have significantly expanded our experimental section to provide deeper analysis and a clearer demonstration of the failure modes of existing solvers.

---

> > ### Public Comment · ~Zander_W._Blasingame1 · 2025-11-27
> >
> > Hi,
> >
> > One of the authors of the Rex scheme [2] here. I read through this paper and the proposed EES schemes are different to what we studied. As the authors pointed out the Rex family of solvers is focused on diffusion/flow models in particular and is extension of the McCallum-Foster method for these models. Thus, the extension of the McCallum-Foster method to SDEs applied to the RDE Runge-Kutta schemes would provide a similar comparison to our Rex (SDE) schemes. For the SDE case we extended the McCallum-Foster method to SDEs and applied them to the Stochastic Runge-Kutta methods proposed in [3].
> >
> > In summary, I think the proposed EES scheme is distinct from the McCallum-Foster method and SDE extension we studied in Rex. Hope this helps clarify things!
> >
> > [3] Foster, James M., Goncalo Dos Reis, and Calum Strange. "High order splitting methods for SDEs satisfying a commutativity condition." SIAM Journal on Numerical Analysis 62.1 (2024): 500-532.

---

### Official Review · Reviewer_F8uU · 2025-11-01

**Soundness:** 3
**Presentation:** 3
**Contribution:** 3
**Rating:** 8
**Confidence:** 2

**Summary:**

This paper extends the Explicit and Effectively Symmetric (EES) schemes proposed by [Shmelev05], which is a Runge-Kutta scheme for ODEs, to neural SDEs and RDEs. The method is motivated by the problem of backpropagation through SDE solvers. There are 3 approaches: (1) stores the full forward graph ("discretise–then–optimise"), which is accurate by expensive; (2) integrates an adjoint/backward SDE ("optimise–then–discretise"), which is cheap by inaccurate; (3) algebraically reversible solvers, which promise both accuracy and cheap compute, but are often unstable (e.g., Reversible Heun). The paper provides a memory‑efficient backprop algorithm through explicit RK solvers using a reverse step and in‑place stage‑wise autodiff (Algorithm 1), and provides stability analysis which shows that the method has similar stability to the classical RK3/RK4 methods. The paper also conducts experiments that compare the proposed EESR(2,5) scheme against Reversible Heun on two neural SDE tasks: (1) a high‑volatility 2D OU process and (2) a GBM calibrated to option prices—reporting. For both tasks, EESR reports lower losses and fewer training epochs (Figs. 3–4).

**Strengths:**

The paper is mathematically rigorous and novel. Extending EES schemes from ODEs to the RDE/SDE is natural but non‑trivial, and the high stability is demonstrated both theoretically and experimentally, which solves the posed question of algebraically reversible SDE solvers being unstable. The two training case studies are well-chosen.

**Weaknesses:**

My main concern is that the baseline is narrow. The comparisons focus almost exclusively on Reversible Heun. The comparison with other common baselines would help. Also, including performance measures like wall‑clock time, function‑evaluation counts, and memory profiles, would substantiate the claimed practical advantages. Moreover, both tasks (2D OU, 1D GBM) are low-dimensional and fairly structured; results on higher‑dimensional NSDEs, stiff drifts, or non‑Gaussian rough drivers would better test generality.

**Questions:**

- Can you quantify gradient error by comparing EESR‑based gradients to a “reference” gradient obtained by storing the forward graph at a much smaller step size?
- Section 5 notes that when stability is not an issue, Reversible Heun uses fewer evaluations per step. Can you characterise regimes where Heun is preferable, and perhaps propose a hybrid scheme that switches to EESR only when instability is detected?

---

> ### Author Response · Authors · 2025-11-20
>
> We thank the reviewer for their time and helpful feedback. We hope that the points below address all the reviewer's concerns.
>
> **Weaknesses:**
>
> * We agree that the original baseline was narrow. As far as we are aware, Reversible Heun is the only available explicit reversible SDE solver. Other reversible schemes such as ALF and McCallum-Foster are ODE methods. To broaden the baselines, we have now adapted the original ODE McCallum–Foster method to the SDE setting by applying their techniques to RDE Runge-Kutta schemes, and we include two variants of this solver (using Euler and Midpoint discretisations) across both experiments. Since ALF has the same stability region as Reversible Heun when applied to ODEs, we have not included it in the baselines. Please see Sections 4.1 and 4.2 in the revised paper.
> * Following your suggestions, we have incorporated runtime as an explicit metric in the revised paper. Regarding function evaluation counts, we have adjusted our examples so that each solver’s step size is chosen to fix the number of drift and diffusion evaluations. This ensures that the solvers have comparable computational budgets, enabling a fairer comparison. Please see Sections 4.1 and 4.2 in the revised paper.
> * Regarding dimensionality of the test problems, we should note that although the OU dynamics are low-dimensional, the latent SDE is 32-dimensional, making this test case high-dimensional for the solver. Additionally, we have now replaced the original option price GBM example with an example of high-dimensional GBM dynamics with stiff drift, where the drift is given by A y_t dt for a matrix A with extreme negative eigenvalues. Please see Section 4.2 in the revised paper.
>
> **Questions:**
> * To quantify the gradient error, we have added the MSE of the gradients of the loss during training for the GBM example, computed against reference gradients of a discretise-then-optimise solution. Please see section 4.2 and Figure 5 of the revised paper.
> *  Regarding a Heun-EES hybrid scheme, the updated examples show that even though EES uses more function evaluations, it allows for a larger step size to be taken without impacting performance.  Since the same performance can be achieved by both schemes with the same number of function evaluations, we expect that a hybrid scheme is not generally necessary.

---

### Meta-Review · Area_Chair_MHME · 2026-01-07

**Summary:**

The submitted manuscript is the extension of the previously developed Explicit and Effectively Symmetric solvers to Stochastic Differential Equations.

The major common concern of the reviewers is the limited empirical study: the lack of comparison across computational budgets, the lack of baselines, and the use of low-dimensional synthetic experiments as the main problem considered. Furthermore, reviewers tYYW and VyCj raised detailed technical concerns regarding the computational cost and the choice of hyperparameters.

**Reviewer Concerns:**

The authors extended the empirical comparison to a more-dimensional (d = 25) synthetic task, added the requested baselines, and provided the methods' runtimes. The method remains untested on real-world applications, and the paper's limited contribution, as raised by G7my, remains unaddressed.

**Reviewer Scores:**

Reviewer VyCj explicitly mentions the intention to raise the score, and Reviewer tYYW mentions that most of their concerns were addressed. However, given the reserved initial evaluation and the rebuttal provided, I do not see a reviewer confidently advocating for the paper's acceptance. Thus, I conclude that after the rebuttal, this paper would be a borderline case.

---

### Decision · Program_Chairs · 2026-01-26

Reject